# Trialkoxysilane-Induced Iridium-Catalyzed *para*-Selective C−H Bond Borylation of Arenes

Guodong Ju[1], Zhibin Huang[1] & Yingsheng Zhao ®[1,2]✉

An ideal approach for the construction of aryl boron compounds is to selectively replace a C−H bond in arenes with a C−B bond, and controlling regioselectivity is one of the most challenging aspects of these transformations. Herein, we report an iridium-catalyzed trialkoxysilane protecting group-assisted regioselective C−H borylation of arenes, including derivatives of benzaldehydes, acetophenones, benzoic acids, benzyl alcohols, phenols, aryl silanes, benzyl silanes, and multi-functionalized aromatic rings are all well tolerated and gave the *para*-selective C−H borylation products in a short time without the requirement of inert gases atmosphere. The site-selective C−H borylation can be adjustable by installing the developed trialkoxysilane protecting group on different functional groups on one aromatic ring. Importantly, the preparation process of the trialkoxychlorosilane is efficient and scalable. Mechanistic and computational studies reveal that the steric hindrance of the trialkoxysilane protecting group plays a key role in dictating the *para*-selectivity.

Highly regioselective direct C−H functionalizations provide an efficient approach for the construction of important synthetic units[1–5]. However, the direct transformation of a specific C−H bond into other functional groups is always a challenging task[6–8]. Various ligand-promoted transition-metal-catalyzed selective C−H functionalizations have been explored in the past two decades[9–11], among which the Ir-catalyzed direct C−H borylation has received attention as a highly atom-economical preparation method for important organoboron reagents[12–15]. Thus, this strategy has immensely promoted the development of organoboron chemistry[16,17]. While Ir-catalyzed *ortho*- or *meta*-selective C−H borylation reactions have been extensively investigated, *para*-selective C−H borylation reactions remain rare[18–20]. This may be attributed to the greater difficulty in using directed templates or ligands to selectively activate the *para*-C−H bonds in aromatic ring[21]. To the best of our knowledge, there are two strategies to accomplish *para*-selective C−H borylation, respectively by utilizing weak interactions or steric hindrance between ligand and substrate (Fig. 1a)[22]. Weak interactions between the ligand and substrate provide an efficient route for *para*-selective C−H borylation.

For example, Nakao and co-workers reported a cooperative Ir/Al catalytic system to perform the *para*-selective C−H borylation of benzamides[23]. Similarly, O--K non-covalent interactions[24], ion-pair ligand-directed interactions[25], and intermolecular hydrogen bond-directed interactions[26–28] were utilized to achieve the *para*-selective C−H borylation of aromatic esters, quaternary ammonium salts, Weinreb amides, and aryl sulfonyl compounds. The groups of Phipps, Smith, and Maleczka developed the *para*-selective C−H borylation of arenes via ion-pairing with bulky countercations[29,30]. This strategy exhibited an excellent substrate scope, including phenols, anilines, benzyl alcohols, and sulfonates. Smith and Maleczka groups controlled the *para*-borylation of aniline through steric and hydrogen bonding[31]. Compared with weak interactions, there are relatively few reports on sterically controlled *para*-selective C−H borylation of arenes. In a pioneering work, Itami et al. reported that a bulky diphosphine ligand could alter the site-selectivity of Ir-catalyzed C−H borylation[32]. In their report, when quaternary carbon-substituted arenes were used, *para*-selective C−H borylations were observed, however, the site-selectivity was usually

---

[1]Key Laboratory of Organic Synthesis of Jiangsu Province, College of Chemistry, Chemical Engineering and Materials Science, Soochow University, Suzhou 215123, China. [2]School of Chemistry and Chemical Engineering, Henan Normal University, Xinxiang 453000, China. ✉e-mail: yszhao@suda.edu.cn

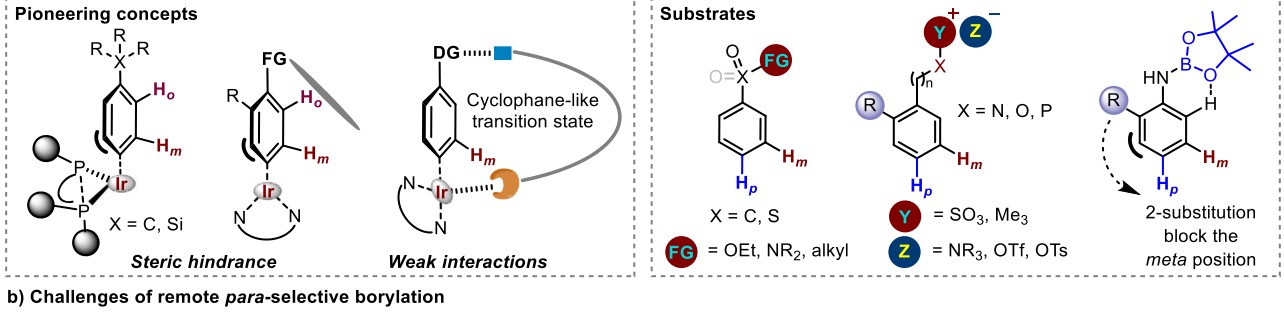

**a)** Summary of pioneering concepts and substrates for *para* C-H borylation

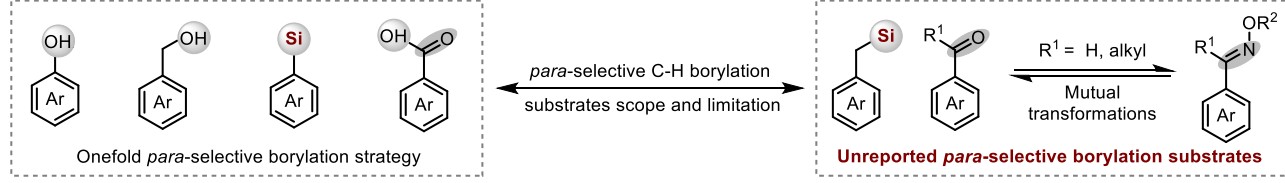

**b)** Challenges of remote *para*-selective borylation

*Challenges: General & practical strategy for para-selective borylation*

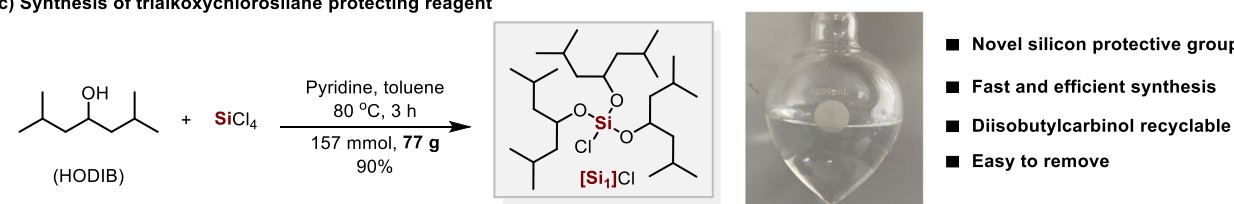

**c)** Synthesis of trialkoxychlorosilane protecting reagent

- Novel silicon protective group
- Fast and efficient synthesis
- Diisobutylcarbinol recyclable
- Easy to remove

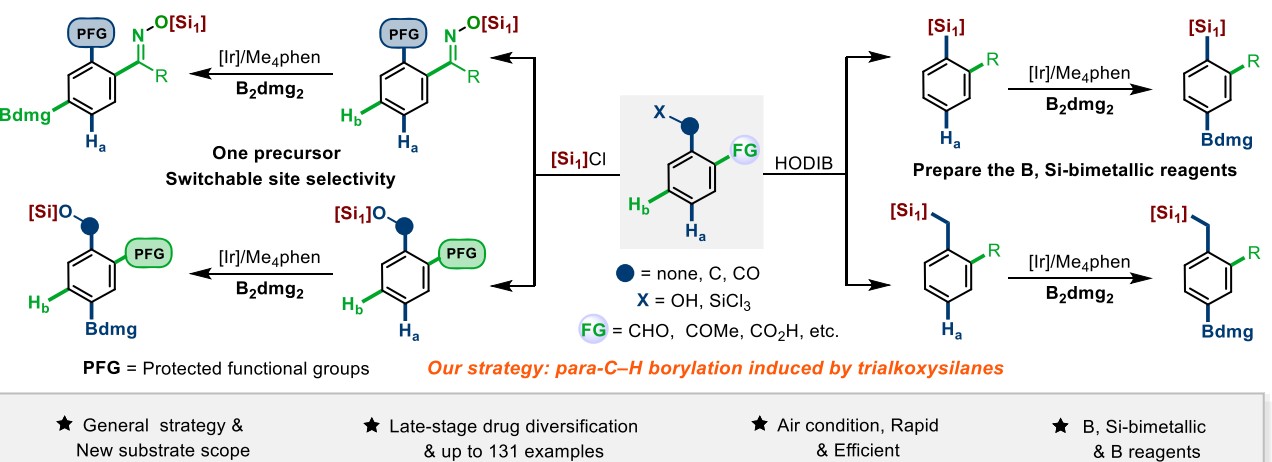

**d)** This work: A practical and general strategy for *para*-selective C-H borylation of arenes

*Our strategy: para-C–H borylation induced by trialkoxysilanes*

★ General strategy & New substrate scope  ★ Late-stage drug diversification & up to 131 examples  ★ Air condition, Rapid & Efficient  ★ B, Si-bimetallic & B reagents

**Fig. 1 | Ir-catalyzed *para*-selective C–H bond borylation of arenes. a** Summary of pioneering concepts and substrates for *para* C-H borylation. **b** Challenges of remote *para*-selective borylation. **c** Synthesis of trialkoxychlorosilane protecting reagent. **d** Trialkoxysilane-induced Ir-catalyzed *para*-selective C–H borylation of arenes. $B_2dmg_2$ = 4,4,4′,4′,6,6,6′,6′-Octamethyl-2,2′-bi(1,3,2-dioxaborinane). [$Si_1$] = Tris((2,6-dimethylheptan-4-yl)oxy)silyl. HODIB = Diisobutylcarbinol.

low. The utilization of ligand-substrate distortion to realize *para*-selective C−H borylation of aromatic amides was also reported by Chattopadhyay group[33]. Subsequently, this research group achieved the control of the boronation *para*-selectivity by designing a ligand framework to exploit the steric crowding generated by the in situ generated N-Bpin and the *ortho*-substitution of the aniline substrate[34]. These developed strategies have greatly extended the substrate scope of C−H borylation and provide a direct approach for the construction of various organoboron reagents. However, ketone, aldehyde oxime derivatives, benzylsilanes, and multi-functionalized substrates have still not been reported for C−H borylation, and the *para*-selective C−H borylation strategies for other reported arene building blocks are relatively onefold, employing

weak interactions in most cases (Fig. 1b). In addition, a series of tedious synthesis steps for non-commercial ligands also limits their application. In medicinal chemistry, emerging regioselective C−H borylation methods offer opportunities to explore the chemical libraries inaccessible by traditional synthesis. Therefore, it is imperative to develop general and readily accessible regioselective borylation reaction methods[35–39].

Silicon-based protecting groups are particularly useful as they can be selectively installed and removed under mild conditions. Importantly, the trialkoxychlorosilane protecting reagents are easily prepared from various alcohols, leading to adjustable steric effects. Since C−H borylation is highly sensitive to the steric effect of substituents, we believe that when aromatic compounds, such as oximes, phenols,

benzyl alcohol, etc., are protected with a bulky trialkoxysilane highly *para*-selective C–H borylation can be achieved with a simple Ir-catalyst system ([Ir(cod)(OMe)]₂/Me₄phen). Along these lines, we synthesized a bulky trialkoxychlorosilane protecting reagent (Fig. 1c). Herein, we demonstrate the implementation of this protocol, developing a site-selective C–H borylation controlled by switching the trialkoxysilane protecting group at different functional groups on the aromatic ring (Fig. 1d). This approach dispenses with the traditional strict anhydrous and oxygen-free conditions, and the reaction is highly fast. Derivatives of benzaldehydes, acetophenones, benzoic acids, benzyl alcohols, and phenols all performed well, offering *para*-selective borylated compounds in good to excellent yields, highlighting the synthetic importance of this method. Aryl and benzyl silanes can also achieve *para*-selective C-H borylation under the induction of trialkoxysilane, providing an efficient approach to various B, Si bimetallic reagents. Moreover, the regioselective C–H late-stage borylation of drugs, including clopidogrel, aspirin, and zaltoprofen has been efficiently completed. We believe that this protocol represents the most general strategy for *para*-selective borylation available to date, achieving *para*-selective C–H borylation of seven types of arenes, and indeed provides an effective complement to existing protocols.

## Results

### Reaction development
To the best of our knowledge, benzaldehyde derivatives were not optimum substrates in previously reported Ir-catalyzed *para*-selective

C–H borylations[40,41]. In this context, we first converted 2-chlorobenzaldehyde to its oxime and further protected it with trialkoxysilanes to investigate the influence of steric effect on site-selectivity in the standard Ir-catalyzed C–H borylation reaction. We treated **1a** (0.2 mmol, 1 equiv) with B₂dmg₂ (1.5 equiv) in the presence of [Ir(cod)(OMe)]₂ (1.5 mol%) and Me₄phen (3.0 mol%) in cyclohexane at 100 °C for 1 hour (Fig. 2). The starting material **1a** was completely transformed, affording the borylation product **2a** a yield of 86%. However, the site-selectivity was rather poor (*p*/others = 4/1). When triethylchlorosilane and triisopropylchlorosilane were tested, the site-selectivity was slightly improved (**2b** and **2c**). When trialkoxychlorosilanes such as trimethoxychlorosilane, triisopropoxychlorosilane, and tricyclohexyloxychlorosilane (**2d**–**2f**) were used as protecting reagents, the site-selectivity was improved with increasing steric hindrance. Next, bulkier trialkoxychlorosilanes were synthesized and used as protecting reagents, which greatly enhanced the site-selectivity. Satisfactory *para*-selective borylation (*p*/others > 20/1, **2i**) could be achieved when using chlorotris((2,6-dimethylheptan-4-yl)oxy)silane (TDBSCl). Other bulky trialkoxysilane-protected 2-chlorobenzaldoxime substrates (**2g** and **2h**) were further tested, none of which offered better yield and selectivity than **2i**. Interestingly, when B₂pin₂ was used instead of B₂dmg₂, the yield and selectivity of the borylated products (**2j**) were poorer than those of B₂dmg₂. Although the reason is unclear, we attributed the reduced site-selectivity to the lower steric effect of B₂pin₂ than that of B₂dmg₂. After determining the suitable trialkoxysilane protecting group, we then evaluated the

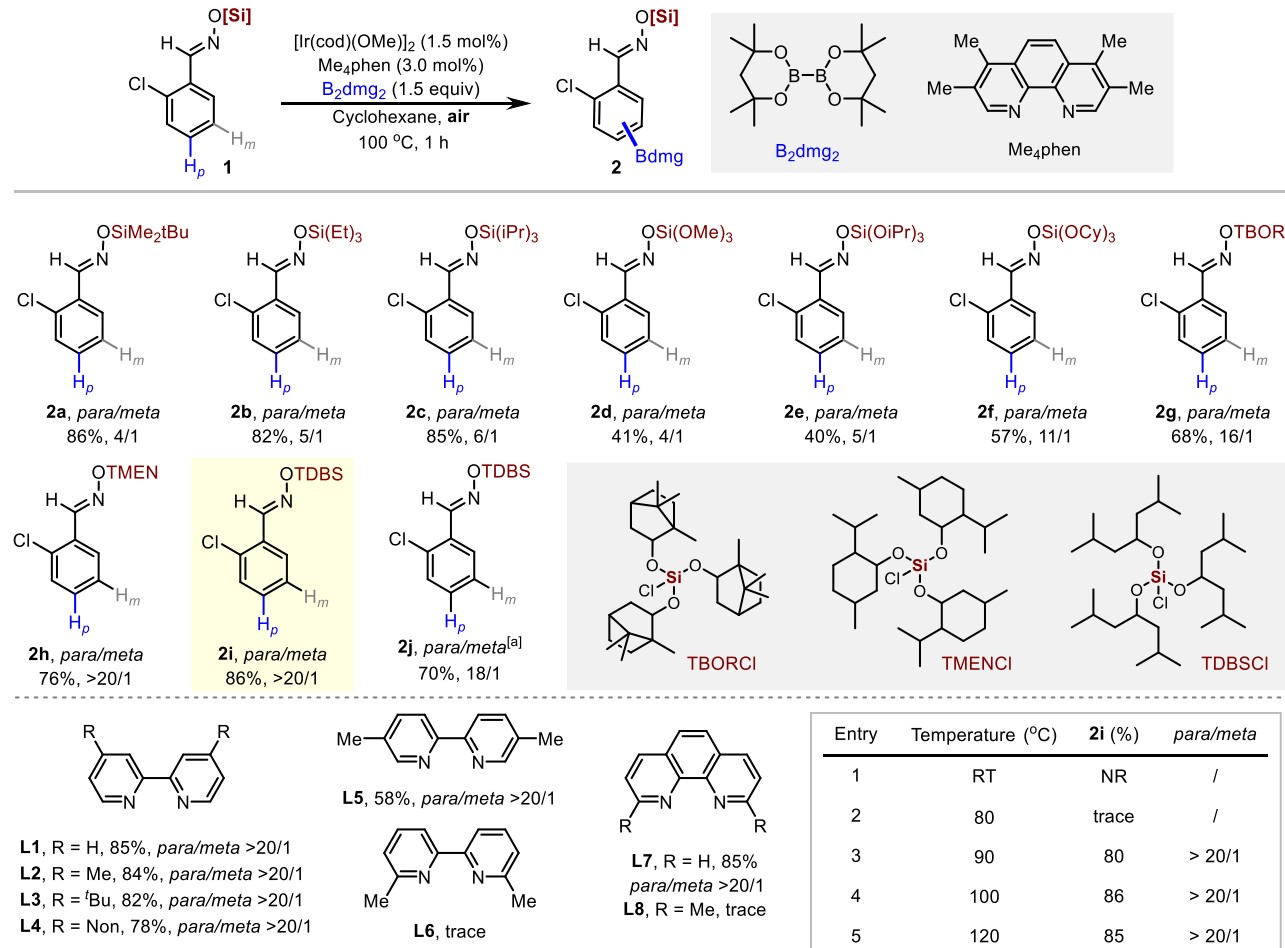

**Fig. 2 | Effects of various silicon functional groups, ligand and temperature for *para* C-H borylation.** Reaction conditions: substrate **1** (0.2 mmol), B₂dmg₂ (1.5 equiv), [Ir(cod)(OMe)]₂ (1.5 mol%), Me₄phen (3.0 mol%), cyclohexane (1 mL), 100 °C, 1 h, isolated yield. Ratios of *meta* to *para* were determined from the crude ¹H-NMR spectra after borylation. [a] B₂pin₂ used instead of B₂dmg₂. B₂dmg₂ = 4,4,4′,4′,6,6,6′,6′-Octamethyl-2,2′-bi(1,3,2-dioxaborinane). Me₄phen = 3,4,7,8-Tetramethyl-1,10-phenanthroline.

effects of ligands and temperature on the reactivity and selectivity. To our delight, except for **L6** and **L8** with high steric hindrance, other commonly commercialized bipyridine (**L1-L5**) and phenanthroline (**L6**) ligands have observed good reactivity and para selectivity. The

excellent performance of numerous ligands further demonstrates the practicality of this strategy. Further evaluation of the reaction conditions was conducted, and the reaction could not occur when the temperature was reduced to 80°C. It is worth mentioning that the

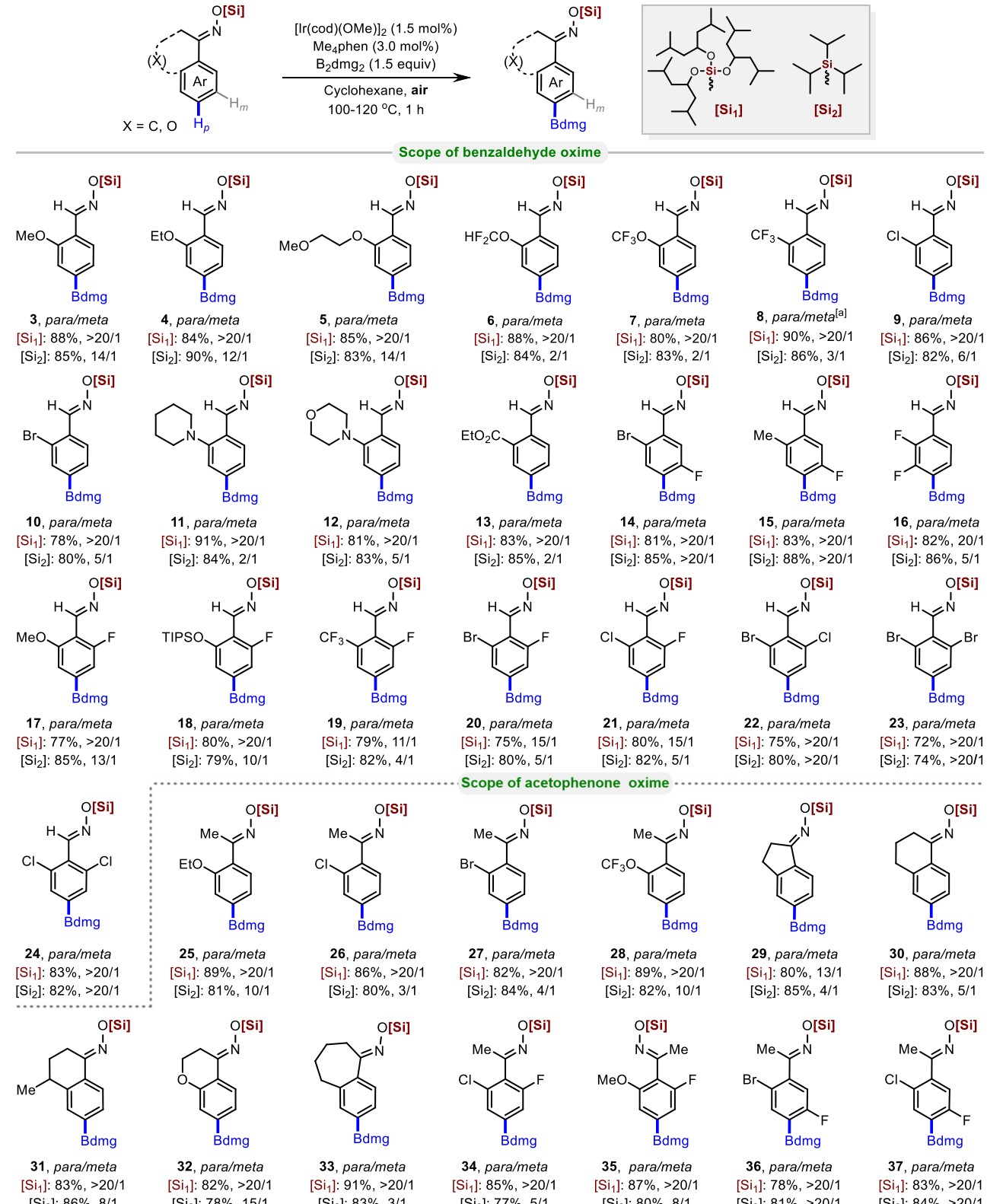

**Fig. 3 | Scope of *para* C−H borylation of benzaldehyde and acetophenone derivatives.** Reaction conditions: Substrate (0.2 mmol), B₂dmg₂ (1.5 equiv), [Ir(cod)(OMe)]₂ (1.5 mol%), Me₄phen (3.0 mol%), cyclohexane (0.2 M), 100–120 °C,

1 h, isolated yield. [a] dtbpy used instead of Me₄phen. Ratios of *meta*- to *para*- were determined from the crude ¹H-NMR spectra after borylation. [Si₁] = Tris((2,6-dimethylheptan-4-yl)oxy)silyl. [Si₂] = Triisopropylsilyl.

catalytic system does not require strict anhydrous and anaerobic conditions. Although the specific reasons are unclear, we may speculate that B$_2$dmg$_2$ may form a cage-like structure with large steric hindrance with [Ir(cod)OMe]$_2$ and Me$_4$phen to protect the iridium catalytic center, thereby act as a barrier between water and oxygen molecules in air and solvent[30,42,43]. In addition, the high reactivity of B$_2$dmg$_2$ greatly increases the reaction rate, leading to the reactions to be completed quickly.

## Substrate scope

With the optimized conditions in hand, various aromatic oximes were evaluated to understand the scope and possible limitations of our protocol. Interestingly, 2-substituted arenes were all well-tolerated irrespective of their electronic nature, and a high degree of *para*-selectivity was observed (Fig. 3). For instance, alkoxy-substituted aromatic aldoximes were well-tolerated, giving >20:1 *para*-selectivity (3–5). Arenes bearing difluoromethoxy, trifluoromethoxy, and trifluoromethyl substituents were also compatible, providing the corresponding products (6–8) in good yields and regioselectivities. Chloride and bromide substituents were well-tolerated, both giving >20:1 *para*-selectivity (9 and 10). Strong electron-donating groups such as piperidine and morpholine substituents gave excellent yields and *para*-selectivity of the desired borylated products (11 and 12). An *ortho*-substituted benzaldoxime with a weak electron-withdrawing group (CO$_2$Et) afforded the desired borylated product (13) with excellent selectivity and reactivity under the present reaction conditions. Remarkably, when triisopropylchlorosilane was used as the protecting reagent for the above 2-substituted benzaldoximes, the borylation reaction occurred in good conversion, but the *para*-selectivity was poor in all cases. This further confirms that the steric hindrance of trialkoxysilane groups can affect the site-selectivity of C−H borylation. The *para*-selectivity was found to be remarkably high for 2,5- and 2,3-disubstituted arenes (14–16). The developed method was also tested for 2,6-disubstituted aromatic aldoxime derivatives. As expected,

various 2,6-disubstituted arenes afforded exclusive *para*-borylation products (17–24). Interestingly, when fluorine atoms were present in the 2,6-disubstituted arenes, the trialkoxysilane protecting group significantly increased the *para*-selectivity compared to the less hindered triisopropylsilane (17–21). Next, we surveyed acetophenone oxime derivatives. *Ortho*-substituted acetophenone oxime derivatives with electron-donating groups (OEt) provided the desired borylated products with excellent regioselectivity (25). *Ortho*-substituted acetophenone oxime derivatives with electron-withdrawing groups (Cl, Br, and OCF$_3$) also exhibited excellent regioselectivity, leading to the *para*-borylated products (26–28). Benzocyclone oxime derivatives, including 1-indanone (29), 1-tetralone (30), 4-methyl-1-tetralone (31), 4-chromanone (32), and 1-benzocyclone heptanone (33) oximes, performed well and yielded the desired products with excellent reactivity and regioselectivity. In the case of 2,5- and 2,6-disubstituted acetophenone oxime derivatives (34-37), the *para*-borylated products were obtained in good yields. Similar to aldoximes, the *para*-selectivity was usually poor for acetophenone oxime derivatives when the less hindered triisopropylsilane was used as a protecting group.

Encouraged by the efficiency of this method for the regioselective borylation of aromatic oximes, we next facilely installed bulky hindered trialkoxysilane protecting groups in oxygen-containing building blocks, such as benzoic acid, benzyl alcohol, and phenol derivatives. (Fig. 4). In 2017, Chattopadhyay and co-workers reported the only case of Ir-catalyzed *para*-selective borylation of aromatic esters using a designed L-shaped ligand24. Using our strategy, good tolerance and high regioselectivity were observed for 2-substituted benzoic acid derivatives, including those with halogen atoms (39 and 40), electron-withdrawing groups (41 and 42), and electron-donating groups on arenes (43). Arenes with 2,3-difluoro substituents have shown good *para*-selectivity (44). Since the *para*-selective borylation of benzoate derivatives is currently only possible through non-covalent interactions, the method we developed is an effective complement to current strategies. In 2019, the Phipps and Smith, Maleczka group reported the

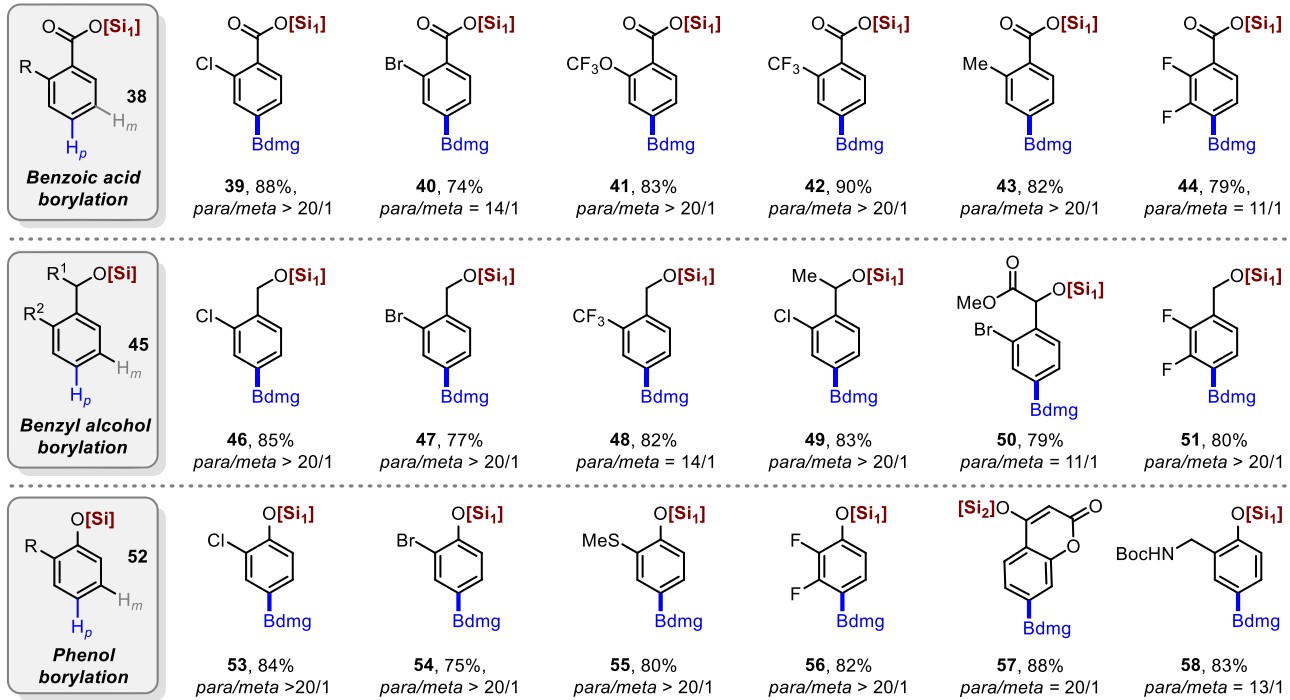

**Fig. 4 | Scope of *para* C−H borylation of benzoic acid, benzyl alcohol and phenol derivatives.** Reaction conditions: Substrate (0.2 mmol), B$_2$dmg$_2$ (1.5 equiv), [Ir(OMe)cod]$_2$ (1.5 mol%), Me$_4$phen (3.0 mol%), cyclohexane (0.2 M), 100–120 °C, 1 h, isolated yield. [Si$_1$] = Tris((2,6-dimethylheptan-4-yl)oxy)silyl. [Si$_2$] = Triisopropylsilyl. Ratios of *meta*- to *para*- were determined from the crude $^1$H-NMR spectra after borylation.

*para*-borylation of phenol and benzyl alcohol derivatives using steric and ion-pair directed strategies[29,30]. Gratifyingly, the installation of bulky sterically-hindered trialkoxysilane protecting groups on benzyl alcohol and phenol derivatives resulted in excellent regioselectivities and yields of their corresponding *para*-borylated products (**46–51** and **53–58**, respectively). The *ortho*-substituted benzyl silicate substrates

performed well, affording the corresponding products with excellent yields and selectivities. For instance, *ortho*-Cl, *ortho*-Br, *ortho*-CF₃, trifluoromethoxy functional groups, and a difluorinated analogue were perfectly compatible with this transformation (**46–51**). Similar results were obtained for phenol derivatives; Phenylsilicates with *ortho*-Cl (**53**), *ortho*-Br (**54**), and *ortho*-SMe (**55**) functional groups, and a

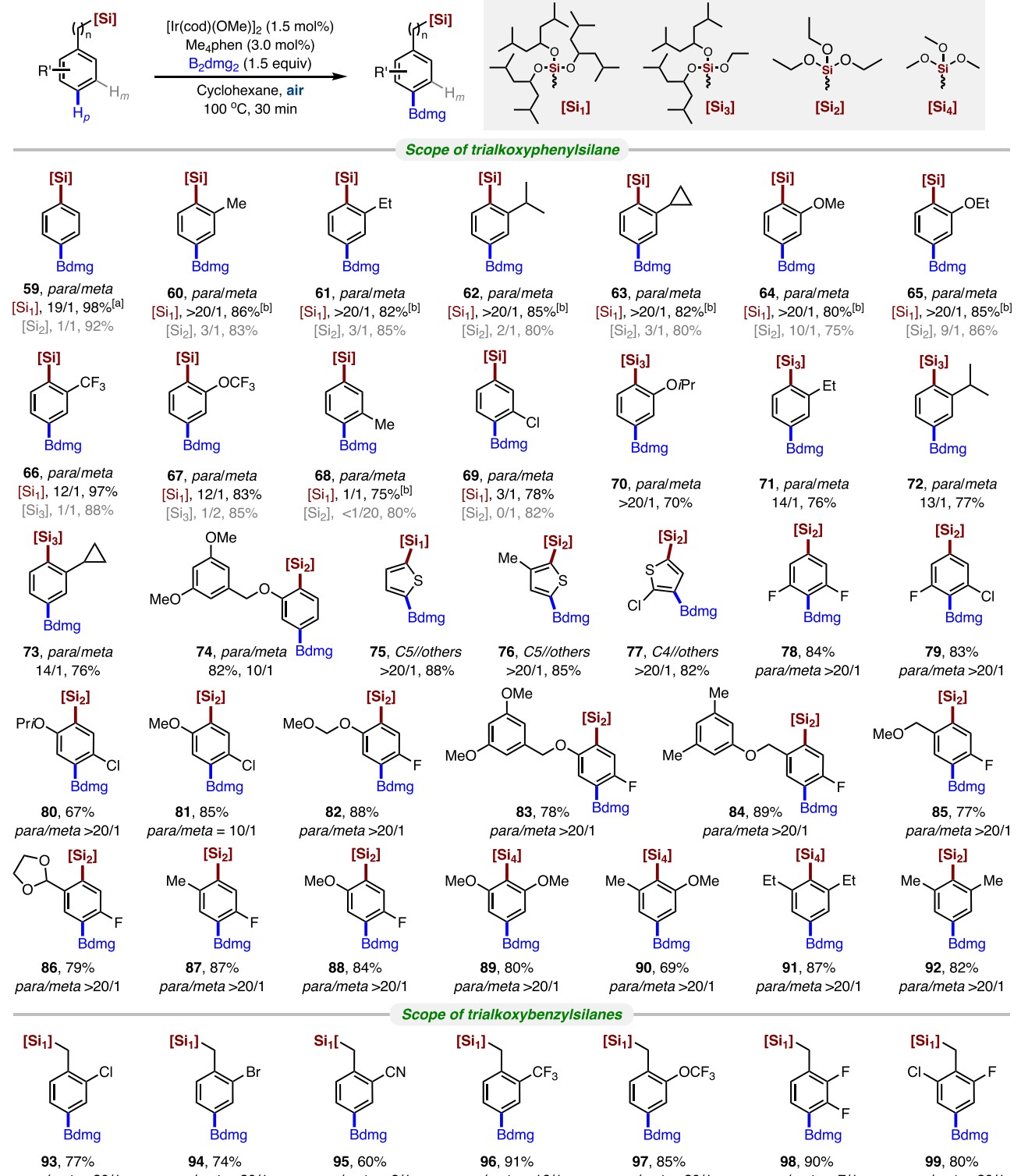

**Fig. 5 | Substrate scope of *para* C–H borylation of arylsilicon compounds.**
Reaction conditions: substrate **3** (0.2 mmol), B₂dmg₂ (0.3 mmol), [Ir(cod)OMe]₂ (1.5 mol%), Me₄phen (3.0 mol%), cyclohexane (0.2 M), 100 °C, 30 min, isolated yield. [a] Reaction scale 3.0 mmol. [b] 4 h. Ratios of *meta* to *para* were determined from the crude ¹H-NMR spectra after borylation.

difluorinated substrate (**56**) gave excellent *para*-selectivity and yields. Remarkably, 4-hydroxycoumarin (dicoumarin intermediate)[44] and 2-hydroxybenzylamine (IsoLG scavenger)[45] were successfully borylated in good yields and regioselectivities (**57** and **58**, respectively). Using commercially available ligands, highly regioselective *para*-borylation of benzoic acid, benzyl alcohol, and phenol derivatives was accomplished, further demonstrating the versatility and practicality of the trialkoxysilane protection strategy.

In all known iridium-catalyzed *para*-selective C-H borylation reactions, the aryl silicon has been less explored. Only three examples describing the bulky diphosphine ligand-enabled *para*-selective C-H borylation have been identified, but the site selectivity was found to be barely satisfactory (*para/others* < 9/1)[32,46]. Aryltrialkoxysilicon is a class of essential compounds that was well applied in the Hiyama cross-coupling reaction[47–50] and material chemistry[51–54]. Also, sterically hindered aryltrialkoxysilicon could be easily prepared from the organohalide reagent, which might be the perfect substrate for performing *para*-selective C-H borylation, thereby generating the useful B, Si-bimetallic reagent[55,56]. We synthesized a variety of different sterically hindered trialkoxyphenyl and benzylsilanes to further examine the functional group tolerance to expand the scope of B, Si bimetallic reagents (Fig. 5). Gratifyingly, the inductive effect of trialkoxysilanes is very versatile for a wide variety of differently substituted phenyl and benzylsilanes, affording high levels of *para*-selectivity and yields of the borated product regardless of the nature of their substituted groups. The yield and site selectivity of the unsubstituted phenylsilane borylation is excellent.

Remarkably, this is a highly selective *para*-position C-H borylation of a monosubstituted substrate that been achieved through the steric hindrance effect of the substrate itself (**59**). The *ortho*-substituted substrates performed well, affording the corresponding products excellent yields and selectivity. For instance, functional groups (methyl, methoxy, trifluoromethyl, and trifluoromethoxy) were fully compatible with this transformation to give the corresponding B-Si bimetallic reagents in excellent yield and selectivity (**60–67**). Also, all the *ortho*-substituted aryltriethoxysilanes gave poor site selectivity. We also explored the C−H borylations of 3-substituted arylsilanes. When using 3-methyl and 3-chlorine substrates, the site selectivity decreased dramatically (**68** and **69**) due to the steric hindrance, which agreed with the results by Hartwig[57,58]. However, its *para*-selectivity significantly improved compared to the triethoxysilane substrate. This further confirms that the steric hindrance of the silyl group can influence the site selectivity of borylation. Since the *para*-selectivity was affected by the alkoxy groups on the silicon, we certified that the substrates containing substituents at the *ortho* position could achieve good site selection if one of the bulky 2,6-dimethyl-4-heptyloxy were substituted with a smaller alkoxy group (**70–73**). However, the site-selectivity was slightly reduced compared to the tridiisobutylmethoxy group substituted silyl group (**61–63** *vs.* **71–73**). A bulky *ortho*-substituted substrate of (3,5-dimethoxyphenyl)methoxy was next examined (**74**), yielding the borylation product in good yield with good site selectivity (*para/meta* = 10:1). Next, considering the importance of various thiophene molecules, we prepared the corresponding

**Fig. 6 | Switchable site-selective C-H borylation.** Reaction conditions: Substrate (0.2 mmol), B$_2$dmg$_2$ (1.5 equiv), [Ir(cod)(OMe)]$_2$ (1.5 mol%), Me$_4$phen (3.0 mol%), cyclohexane (0.2 M), 100–120 °C, 1 h, isolated yield. Ratios of *meta*- to *para*- were determined from the crude $^1$H-NMR spectra after borylation. [Si$_1$] = Tris((2,6-dimethylheptan-4-yl)oxy)silyl.

2-trialkoxysilylthiophenes. Fortunately, the borylations occurred at C5 (**75** and **76**) or C4 (**77**) with excellent selectivity. Silane could easily block the *ortho* position due to its steric hindrance. At the same time, the other functional group could affect the meta position, leading to remarkable *para* selectivity. Herein, the 3,5 and 2,5-disubstituted aryltriethoxysilanes might be good substrates for this transformation. As expected, good *para*-selective C–H borylation reactions were achieved in all cases (**78–88**). In additionally, 2,6-disubstituted aryl silanes were also compatible, providing the *para*-borylated products in good yields (**89–92**). The developed method was also tested against trialkoxybenzylsilanes, which provided satisfactory *para*-selective borylation. Trialkoxybenzylsilanes with *ortho*-chloro and *ortho*-bromo substituents on the phenyl ring furnished **93** and **94** in good yield and excellent *para*-selectivity, providing the opportunity for subsequent selective cross-coupling reactions with two different handles. Trialkoxybenzylsilanes bearing electron-withdrawing substituents such as nitrile (**95**), trifluoromethyl (**96**), and trifluoromethoxy (**97**) groups were also found to be compatible, giving a high level of *para*-selectivity. As expected, 2,3-disubstituted and 2,6-disubstituted trialkoxybenzylsilanes (**98** and **99**) provided borylation products with good *para*-selectivity.

Among the existing reports on Ir-catalyzed distal borylation reactions of arene and heteroarene substrates, quite a few elegant methods explore the *para*- and *meta*-C–H bond borylation reactions[19–24]. However, using these previously reported methods, it is difficult to achieve *para*- or *meta*-tunable C–H borylation on a substrate. In the current work, we achieved tunable borylation reactions between a variety of substrates, including o-acetylphenol (**100**), salicylaldehyde (**105**), 2-carboxybenzaldehyde (**110**), and salicylic acid (**115**) (Fig. 6). For example, after protecting the individual functional groups of *ortho*-acetylphenol by different routes, the 2-(2-methyl-1,3-dioxolan-2-yl)phenol (**101**) and 1-(2-methoxyphenyl)ethan-1-one oxime (**103**) substrates delivered the desired *para*-

borylated products (**102** and **104**). Likewise, for salicylaldehyde (**105**) and salicylic acid (**110**), after sequential protection of their functional groups, the products 2-(1,3-dioxolan-2-yl)phenol (**106**), 2-((triisopropylsilyl)oxy)benzaldehyde oxime (**108**), 2-hydroxy-N,N-dimethylbenzamide (**111**) and 2-methoxybenzoic acid (**113**) were obtained. Next, after installing bulky trialkoxysilanes, the corresponding *para*-borylation products were furnished under the borylation conditions (**107** and **109**, **112** and **114**). Finally, we carried out the corresponding transformation of 2-carboxybenzaldehyde, and *para*-borylated product of 2-formaldoxime ether-benzoic (**117**) and 2-ester-benzaldehyde oxime acid (**119**) could be obtained in good yields and selectivity.

## Synthetic application
During the development of new drugs and the discovery of drug-like molecules, it is imperative to consider the late-stage modification and functionalization of existing drug molecules[59–61]. However, selectively functionalizing specific C–H bonds of complex molecules is difficult due to the presence of numerous similar C–H bonds[62–64]. To date, there have been no studies on the *para*-selective borylation of benzoic acid, benzyl alcohol, and acetophenone-based drug molecules. Gratifyingly, the late-stage borylation of various bioactive molecules and drugs was successfully achieved through our strategy (Fig. 7). For example, the benzoic acid-based compound aspirin[65] has anti-inflammatory and analgesic properties, and the aromatic ketone-based compound zaltoprofen[66] is a non-steroidal anti-inflammatory drug (Fig. 7a). These drugs exhibit good regioselectivities and yields in borylation reactions after the installation of bulky silicon-protecting groups (**120** and **121**). Benzyl alcohol-based bioactive compounds such as 2-chlorcyclomandelate[67] were found to display excellent borylation selectivity (**122**). Delightfully, our methodology was successfully applied to the late-stage borylation of clopidogrel through an efficient multistep pathway (Fig. 7b). Clopidogrel is a platelet aggregation inhibitor with a high market share, and (*R*)-o-chloromandelate (**123**) is a

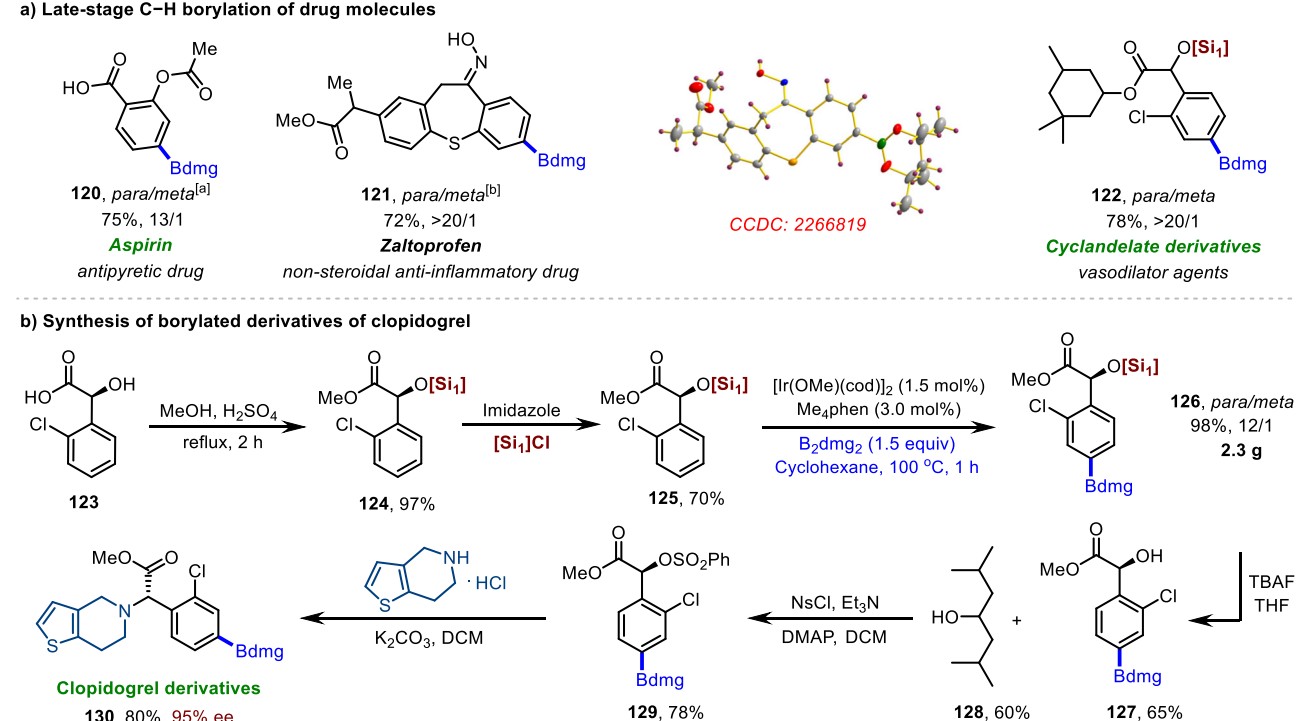

**Fig. 7 | Late-stage C–H borylation of pharmaceutically important molecules. a** Late-stage C–H borylation of drug molecules. [a] Borylation then DMF/H₂O. [b] Borylation then TBFA/THF. **b** Synthesis of borylated derivatives of clopidogrel. [Si] = Tris((2,6-dimethylheptan-4-yl)oxy)silyl.

key intermediate for the preparation of clopidogrel[68]. Starting from **123**, we synthesized the substrate **125** after esterification and silylation. To demonstrate the synthetic utility of our strategy, we performed a gram-scale synthesis that delivered the desired product **126** with

excellent yield and good regioselectivity. Subsequent deprotection of **126** yielded product **127** while recovering the starting material diiso-butylcarbinol (**128**) in moderate yields, further demonstrating the practicability of this strategy. Finally, the late-stage borylation

**Fig. 8 | Silylarenes as a useful building block for constructing functional compounds.** Yields reported are for isolated materials. [Si₁] = Tris((2,6-dimethyl-heptan-4-yl)oxy)silyl. [Si₂] = Triisopropylsilyl. Reaction conditions: (i) 3.0 equiv ICl, CH₂Cl₂, 23 °C, 16 h. (ii) 5.0 mol% Pd(PPh₃)₄, 1.5 equiv TBAF, 5.0 mol% H₂O, 1.5 equiv CuI, 80 °C, 18 h. (iii) 0.5 equiv benzimidazole, 1.1 equiv Cu(OAc)₂, 1.5 equiv TBAF,

DMF, 23 °C, 36 h. (iv) 1.5 equiv Br₂, CH₂Cl₂, 0 °C, 2 h. (v) 2.5 mol% Pd(dba)₂, 2.0 equiv TBAF, THF, 55 °C, 14 h. (vi) 5.0 mol% [Ir(cod)OMe]₂, 1.5 equiv TBAF, toluene/H₂O (6:1), 120 °C, 24 h. (vii) 2.0 equiv Selectfluor, 2.0 equiv Ag₂O, 1.1 equiv BaO, acetone, 90 °C, 2 h.

synthesis of (*S*)- clopidogrel was also carried out successfully according to the literature procedure (**129** and **130**)[69].

The silyl-substituted arene can be easily transformed in the presence of the Si-O bonds, which would facilitate the fast construction of various bioactive molecules, further highlighting the importance of these B, Si-bimetallic reagents (Fig. 8). Felbinac is well-known for its anti-inflammatory and analgesic effects. A silyl-substituted Felbinacetyl (**131**) was readily prepared from **59a** in one pot through a palladium-catalyzed Suzuki coupling (Fig. 8a). The silyl group further undergoes a cross-coupling reaction to install an aryl group on the Felbinacetyl. Various functional groups can be efficiently introduced, including nitrogen heterocycle, alkene, allyl, bromide, and iodine (**133–139**)[70–73]. A fluorine group was also

successfully introduced to Felbinacetyl (**140**)[74]. Through sequential transformation, the B, Si-bimetallic reagent **88** can be converted into a fluorine-containing biphenylacetic silyl derivative **141**, and the iodine group **142** is further introduced under mild conditions[75] (Fig. 8b). Using the B, Si-bimetallic reagents **58** and **92**, halogenation and oxidation can be carried out sequentially at the positions of C-B and C-Si bonds, depending on the chosen reagents and conditions (**143-146**), further demonstrating the orthogonal reactivity of the synthon (Fig. 8c).

## Mechanistic investigation

To demonstrate the utility of this *para*-C−H borylation, we have shown that the Bdmg introduced by 2-chloromandelic acid (an

**Fig. 9 | Synthetic application.** Yields reported are for isolated materials. **a** Derivatization of C-H borylation products of 2-chloromandelic acid derivatives. **b, c** Synthetic transformation and deprotection. [Si$_1$] = Tris((2,6-dimethylheptan-4-yl)oxy)silyl. Reaction conditions: (i) 2.5 mol% Pd(PPh$_3$)$_4$, 2.0 equiv K$_2$CO$_3$, 1.1 equiv Ar-Br, dimethoxyethane/H$_2$O, 80 °C, 12 h. (ii) 5.0 mol% Pd(dppf)Cl$_2$, 3.0 equiv K$_2$CO$_3$, 1.1 equiv Ar-I, THF/H$_2$O = 4/1, 60 °C, 12 h. (iii) 5.0 mol% Pd(dppf)Cl$_2$, 3.0 equiv K$_2$CO$_3$, 1.1 equiv Ar-I, THF/H$_2$O = 4/1, 60 °C, 12 h. (iv) 4.0 mol% Pd(PPh$_3$)$_4$, 10.0 equiv Na$_2$CO$_3$, 1.2 equiv RBr, tolueneEtOHH$_2$O (10:5:1), 80 °C, 13 h. (v) 1.0 mol % Pd$_2$(dba)$_3$.CHCl$_3$, 4.0 mol % PPh$_3$, 4.0 equiv K$_2$CO$_3$, 1.2 equiv BnBr, (10/1) THF/H$_2$O, 100 °C, 24 h. (vi) 3.0 equiv H$_2$O$_2$, NaOH (2.0 M), 0 °C to rt, 2 h. (vii) 20 mol% Cu(OAc)$_2$, 2.0 equiv B(OH)$_3$, 4 Å M.S., MeCN, O$_2$, 85 °C, 24 h. (viii) 2.0 mol% [Ir(cod)OMe]$_2$, THF, 10.0 equiv D$_2$O, 80 °C, 12 h.

intermediate of clopidogrel) could be converted into various functional groups under mild reaction conditions (Fig. 9a). Each derivative was obtained from **143** in a single step, e.g., a Pd-catalyzed Suzuki Miyaura cross-coupling reaction to obtain arene (**148**) and heteroarenes (**149** and **150**)[76], as well as alkynylated (**151**)[77], and benzylated (**152**)[78] derivatives. Phenol derivatives (**153**)[79] can be produced under oxidation with hydrogen peroxide, while the diarylamine compound (**154**)[80] was prepared by a Cu-catalyzed Chan-Evans-Lam coupling reaction. The deuterated compound (**155**)[81] was readily obtained from **143** by converting Bdmg into deuterium with $D_2O$ under Ir-catalysis. Further indicating the synthetic application of the protocol, the substrate **58a** derived from the conversion of 2-hydroxybenzylamine underwent a gram-scale reaction (**58**) and subsequent deprotection (**156**), resulting in good yield and selectivity (Fig. 9b). To our delight, Boc and silyloxy group can be easily removed simultaneously in the presence of trifluoroacetic acid. Similarly, to demonstrate the advantages of synthetic robustness and easy removal with silicon-protecting group, we synthesized the biaryl (**157**) in a one-pot method via borylation and sequential Suzuki cross-couplings (Fig. 9c). Later, the trialkoxysilane protecting group was removed in the presence of formaldehyde and hydrochloric acid to finally yield the 4-acetylbiphenyl derivative (**158**)[82]. Since the direct transformation of remote C−H bonds remains a considerable challenge, these derivatization reactions demonstrate the importance and utility of our methodology.

The steric contour plots (Fig. 10a), van der Waals surfaces (Fig. 10b), and lowest free energy conformation geometries (Fig. 10c) based on DFT geometry optimization were explored to illustrate the steric environment of the compounds **1c**, **1h**, and **1i**. The percentage of buried volume of **1c** (%$V_{bur}$ = 72.6), **1 h** (%$V_{bur}$ = 75.2), and **1i** (%$V_{bur}$ = 80.9) increased sequentially[83,84]. These results indicated that the bulky silicon group sterically crowded the *ortho-* and *meta-*positions but allowed the *para-*position to be exposed, thereby enabling the *para-*selective C−H borylation reaction[85,86]. DFT calculation has been performed to explain the origin for the different reactivity of substrate **1c**, **1h**, and **1i** (see Supporting Information for computational method). As shown in Fig. 10d, The Ir(V) complexes for three substrates in the catalytic cycle was depicted, and they would proceed to the further reductive elimination step. In complex **1c**, the substate **1c** possessed relatively small *i*Pr substituent and it could decrease the steric hinderance between *i*Pr and Bdmg group, resulting in a more feasible reductive elimination step. However, for complex **1h** and **1i**, the relatively large and bulky substituents could increase the steric hinderance between substrate and Bdmg group, which would suppress the elimination step. Therefore, substrates **1h** and **1i** were less reactive compared with substrate **1c**.

## Discussion

In conclusion, we have developed a trialkoxysilane steric effect-induced iridium-catalyzed *para-*selective C−H borylation of various arene building blocks. Remarkably, a boronate group was introduced into the *para-*position of benzaldehyde and acetophenone derivatives. Simultaneously, it is highly efficient for the *para-*selective borylation of benzoic acid, benzyl alcohol, and phenol derivatives, thereby enriching the synthetic toolbox for regioselective C−H borylation. Interestingly, the steric induction effect of

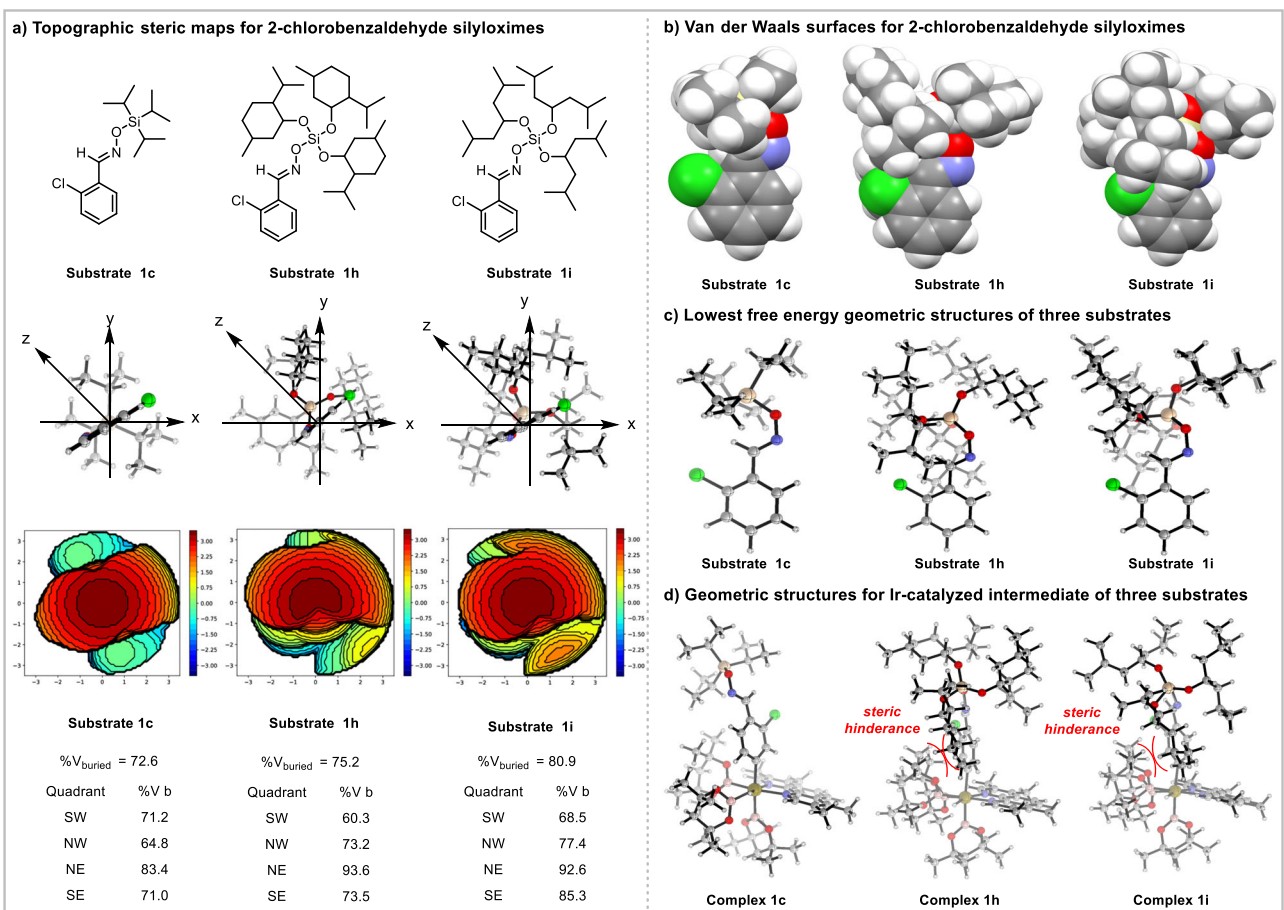

**Fig. 10 | Computational mechanistic study. a** Topographic steric maps for 2-chlorobenzaldehyde silyloximes. **b** Van der Waals surfaces for 2-chlorobenzaldehyde silyloximes. **c** Lowest free energy geometric structures of three substrates. **d** Geometric structures for Ir-catalyzed intermediate of three substrates. %$V_{Bur}$ = percentage of buried volume.

trialkoxysilanes can also realize the *para*-selective borylation of phenyl and benzyl silanes to prepare various B,Si bimetallic reagents. The sequential transformations were successfully applied in the decoration of bioactive compounds. Trialkoxysilanes can be removed easily and the corresponding alcohols can be recovered during the post-processing, thus improving the atom economy. The developed method shows a broad substrate scope, functional group tolerance, and excellent reactivity, as most of the reactions were completed efficiently within 1 hour. This method provides a practical and valuable route to late-stage borylation of several well-known drugs, including clopidogrel, aspirin, and zaltoprofen. The basis for the *para*-selectivity is the steric crowding effect produced by the trialkoxysilane functionality, which effectively blocks the *ortho*- and *meta*-position C–H bonds. This truly general and predictable strategy for site-selective C–H borylation may apply to a range of aromatic remote C–H functionalization reactions, and related studies are currently underway in our laboratory.

## Methods

### General procedure for para-selective C-H borylation of arenes

A mixture of substrate (0.2 mmol, 1.0 equiv), $B_2dmg_2$ (84.6 mg, 0.3 mmol, 1.5 equiv), [Ir(cod)OMe]$_2$ (2.0 mg, 0.003 mmol), Me$_4$phen (1.5 mg, 0.006 mmol), and cyclohexane (1.0 mL) were added to a 15 mL glass vial under air atmosphere. The glass vial was capped with a teflon pressure cap and placed into an aluminum block pre-heated to 100 °C for 30 min. After completion, cyclohexane was removed under reduced pressure and chromatographic separation with silica gel (20% ethyl acetate in hexane as eluent) gave the borylated product. The *para:meta* ratio of products was reported from the analysis of [1]H NMR. Full experimental details and characterization of new compounds can be found in the Supplementary Methods.

## Data availability

Crystallographic data for the structures reported in this Article have been deposited in the Cambridge Crystallographic Data Centre with numbers CCDC 2266819 (**121**). The authors declare that all other data supporting the findings of this study are available within the article and Supplementary Information files, and also are available from the corresponding author upon request.

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

## Acknowledgements

This work was supported by the Natural Science Foundation of China (No. 22171197), the Major Basic Research Project of the Natural Science Foundation of Jiangsu Higher Education Institutions (21KJA150002), the National Local Joint Engineering Laboratory to Functional Adsorption Material Technology for Environmental Protection (SDGC2121), and the PAPD Project. The project was also supported by the Open Research Fund of the School of Chemistry and Chemical Engineering, Henan Normal University.

## Author contributions

G. J. and H. Z. performed the experiments and DFT calculations and analyzed the data. Z. Y. directed the projects and Z. Y. and G. J.co-wrote the manuscript. All authors contributed to the discussion.

## Competing interests

The authors declare no competing interests.
