## [Peer Review File · Nature Communications]

Reviewers' Comments:

Reviewer #1:

Remarks to the Author:

This manuscript presents a protocol for achieving para-selective C-H borylation of arenes using a sterically-hindered trialkoxysilane reagent. Although Phipps, Smith, and Maleczka et al. had made some progress through a similar strategy, this protocol is applicable to a wider range of substrates and provides good to excellent yields. Additionally, the reaction can be conducted under air conditions and the trialkoxysilane group can be easily removed. The authors have demonstrated the utility of the B,Si-bimetallic reagent through various transformations. Therefore, this work is highly recommended for publication in Nature Communications with minor revisions:

1. According to Product 8 in Table 2, it is feasible to use dtbpy ligands. Therefore, it raises the question of whether it is necessary to utilize Me4Phen ligands in this work. In order to demonstrate the rationality and necessity of the ligands used in this method, the authors should show a Table for screening ligands. Furthermore, the remarkably fast reaction rate raises questions regarding the choice of reaction temperature. Therefore, it would be helpful if the author could provide information on the screening of temperature.

2. The method proposed by the author provides an effective approach to the synthesis of bimetallic reagents with both boron and silicon. Its practicality has been validated through a series of synthetic applications, including sequential boron conversion and silicon conversion. However, to further demonstrate the potential of this method, we suggest investigating the possibility of performing silicon conversion before boron conversion. This would demonstrate the orthogonal reactivity of the synthon.

3. Figure 2 presents a steric map to analyze the steric hindrance caused by various silicon reagents. To make the manuscript more comprehensible, the authors can annotate coordinate axis and its direction, along with a listing of the steric hindrance situation in different directions (bottom left, top left, etc., corresponding to the four quadrants of the steric map). Additionally, solely analyzing the steric hindrance of the substrate appears insufficient in effectively explaining the reaction's selectivity. Therefore, we recommend the authors to take the active catalyst into consideration when elucidating the selectivity of the reaction via computational methods.

4. The methods developed by the authors are efficient and practical, particularly in situations where water and oxygen are present. This is impressive, unlike most iridium chemistry. Therefore, the authors should explain this in the main text.

5. References 6 and 12 are the same, while references 14 and 21 are duplicated. The citations 15 and 17 on line 26 do not align with the context of the article. The citation 31 in line 84 does not correspond to the content of the article. The reference 31 in line 144 should be updated to reference 28. The author of line 149 should be revised to Itami and Chattopadhyay. Quote 27 on line 171 also does not align with the context here.

6. The substrate number in line 221 of the main text should be changed from 102 to 101 to match the number in Table 5.

7. In the spectrum annotation on page 5 of SI, "meat" should be corrected to "meta". The NMR data for substrates 1a, 1b, 30a, 45f, and products 3, 4, etc. contain errors. Please thoroughly review all data and descriptions in the SI to ensure the accuracy of this research.

Reviewer #2:

Remarks to the Author:

The manuscript describes para-selective Ir-catalyzed C-H borylation of arenes bearing bulky silyl protecting groups. The work allows the control of para-selectivity across a wide range of substrates including silyl oximes from aldehydes and ketones, silyl esters of carboxylic acids, silyl ethers of benzyl alcohols and phenols, and aryl silanes, taking advantage of steric hindrance. While the work is highly valuable in the context of organic synthesis, it lacks conceptual novelty as it relies on the strategy of steric hindrance between the Ir catalyst and substrate substituents pioneered by Itami et al. As such, the paper represents a logical extension and improvement of the known methods and may not be considered highly significant in the broader scientific context. Therefore, the reviewer recommends its publication in a specialized journal.

Reviewer #3:

Remarks to the Author:

This manuscript reports on the development of para-Selective C–H Bond Borylation of Arenes by Trialkoxysilane-Induced Iridium-Catalyzed. An ideal approach for the construction of aryl boron compounds is to selectively replace a C–H bond in arenes with a C–B bond. However, controlling regioselectivity is one of the most challenging aspects of these transformations due to subtle differences in the reactivity of diverse C–H bonds. This novel trialkoxysilane reagent enables the iridium-catalyzed selective para-C-H boronization of various functionalized aromatic rings without inert gas protection. Some basic aromatic ketone and aromatic aldehyde derivatives can also be successfully applied to the reaction. Para-selective C–H borylation of aryl and benzyl silanes was also achieved, leading to various B,Si bimetallic reagents. Through their strategy, the authors achieved a series of late boronic modifications of drugs, demonstrating the application of their reactions. Mechanistic and computational studies revealed that the steric hindrance of the trialkoxysilane protecting group plays a key role in dictating the para-selectivity. Accordingly, I am in favor of publication in Nature Communications upon a couple of issues that need to be addressed.

1. In Figure 1b, the challenge described by the authors is that of para-C-H borylation of aromatic ketones, aromatic aldehydes, however, there is an ambiguity in expression in terms of the authors' substrate studies using derivatives of aromatic ketones and directional aldehydes.
2. The condition screening is suggested to be added in supplementary materials
3. Boc protecting group should be deprotected to form free amine on Table 3.
4. The authors should have tried the substrate part with a heterocyclic compound.

First of all, we acknowledge the two reviewers for their valuable comments they made to the title manuscript. The point-by-point responses to the editor and reviewer's comments are listed *in italic text in blue*.

Reviewer #1 (Remarks to the Author):

This manuscript presents a protocol for achieving *para*-selective C-H borylation of arenes using a sterically-hindered trialkoxysilane reagent. Although Phipps, Smith, and Maleczka et al. had made some progress through a similar strategy, this protocol is applicable to a wider range of substrates and provides good to excellent yields. Additionally, the reaction can be conducted under air conditions and the trialkoxysilane group can be easily removed. The authors have demonstrated the utility of the B,Si-bimetallic reagent through various transformations. Therefore, this work is highly recommended for publication in Nature Communications with minor revisions:

Response: We thank the reviewer for the positive comments and the support of publication of our work in Nat. Commun. after addressing the issues.

(1) According to Product 8 in Table 2, it is feasible to use dtbpy ligands. Therefore, it raises the question of whether it is necessary to utilize Me₄Phen ligands in this work. In order to demonstrate the rationality and necessity of the ligands used in this method, the authors should show a Table for screening ligands. Furthermore, the remarkably fast reaction rate raises questions regarding the choice of reaction temperature. Therefore, it would be helpful if the author could provide information on the screening of temperature.

Response: We thank the reviewer's valuable advice. We have added a table for ligand and temperature screening in Table 1 and supplemented relevant content in the main text. Me₄phen is one of the classic ligands for iridium catalyzed C-H borylation. We consider that compared to other ligands, Me₄phen has a larger steric hindrance effect and better universality in substrate expansion. Therefore, we choose to use Me₄phen as the ligand.

			Entry	Temperature (°C)	2i (%)	para/meta
L1, R = H, 85%, para/meta >20/1	L5, 58%, para/meta >20/1	L7, R = H, 85% para/meta >20/1	1	RT	NR	/
L2, R = Me, 84%, para/meta >20/1	L6, trace	L8, R = Me, trace	2	80	trace	/
L3, R = ^t Bu, 82%, para/meta >20/1			3	90	80	> 20/1
L4, R = Non, 78%, para/meta >20/1			4	100	86	> 20/1
			5	120	85	> 20/1

*“After determining the suitable trialkoxysilane protecting group, we then evaluated the effects of ligands and temperature on the reactivity and selectivity. To our delight, except for L6 and L8 with high steric hindrance, other commonly commercialized bipyridine (L1-L5) and phenanthroline (L6) ligands have observed good reactivity and *para* selectivity. The excellent performance of numerous ligands further demonstrates the practicality of this strategy. Further evaluation of the reaction conditions were conducted, and the reaction could not occur when the*

temperature was reduced to 80°C.”

(2) The method proposed by the author provides an effective approach to the synthesis of bimetallic reagents with both boron and silicon. Its practicality has been validated through a series of synthetic applications, including sequential boron conversion and silicon conversion. However, to further demonstrate the potential of this method, we suggest investigating the possibility of performing silicon conversion before boron conversion. This would demonstrate the orthogonal reactivity of the synthon.

Response: We thank the reviewer’s valuable advice. We have added the application of silicon conversion before boron conversion in the revised manuscript.

“Using the B, Si-bimetallic reagents **58** and **92**, halogenation and oxidation can be carried out sequentially at the positions of C-B and C-Si bonds, depending on the chosen reagents and conditions (**143-146**), further demonstrating the orthogonal reactivity of the synthon (Table 7C).”

(3) Figure 2 presents a steric map to analyze the steric hindrance caused by various silicon reagents. To make the manuscript more comprehensible, the authors can annotate coordinate axis and its direction, along with a listing of the steric hindrance situation in different directions (bottom left, top left, etc., corresponding to the four quadrants of the steric map). Additionally, solely analyzing the steric hindrance of the substrate appears insufficient in effectively explaining the reaction's selectivity. Therefore, we recommend the authors to take the active catalyst into consideration when elucidating the selectivity of the reaction via computational methods.

Response: We thank the reviewer’s valuable advice. The detailed information of steric map has been provided, please see Figure 2A. A more comprehensive analysis for the Ir-catalyzed complexes has been performed and the results were given in Figure 2D in the manuscript, the steric hindrance could be observed between the substrates **1h** and **1i** and Bdmg group around the metal center, thereby suppressing the reactivity for substrates **1h** and **1i**.

“DFT calculation has been performed to explain the origin for the different reactivity of substrate **1c**, **1h**, and **1i** (see Supporting Information for computational method). As shown in Figure 2D, The Ir(V) complexes for three substrates in the catalytic cycle was depicted, and they would proceed to the further reductive elimination step. In complex **1c**, the substrate **1c** possessed relatively small *i*Pr substituent and it could decrease the steric hindrance between *i*Pr and Bdmg group, resulting in a more feasible reductive elimination step. However, for complex **1h** and **1i**, the

relatively large and bulky substituents could increase the steric hindrance between substrate and *Bdmg* group, which would suppress the elimination step. Therefore, substrates **1h** and **1i** were less reactive compared with substrate **1c**.”

A) Geometric structures for three molecules

B) Van der Waals surfaces for 2-chlorobenzaldehyde silyloximes

C) Lowest free energy geometric structures for 1c, 1h and 1i

D) Geometric structures for Ir-catalyzed intermediate of three substrates

(4) The methods developed by the authors are efficient and practical, particularly in situations where water and oxygen are present. This is impressive, unlike most iridium chemistry. Therefore, the authors should explain this in the main text.

Response: Thanks for the reviewer’s advice. We have added corresponding explanations in the main text.

“Although the specific reasons are unclear, we may speculate that B_2dmg_2 may form a cage-like structure with large steric hindrance with $[Ir(cod)OMe]_2$ and Me_4phen to protect the iridium catalytic center, thereby act as a barrier between water and oxygen molecules in air and solvent^{30,42,43}. In addition, the high reactivity of B_2dmg_2 greatly increases the reaction rate, leading to the reactions to be completed quickly.”

(5) References 6 and 12 are the same, while references 14 and 21 are duplicated. The citations 15 and 17 on line 26 do not align with the context of the article. The citation 31 in line 84 does not correspond to the content of the article. The reference 31 in line 144 should be updated to reference 28. The author of line 149 should be revised to Itami and Chattopadhyay. Quote 27 on line 171 also does not align with the context here.

Response: Thanks for the reviewers for taking precious time to check the manuscript. We have corrected all the mistakes.

References 6 and 12 are the same, while references 14 and 21 are duplicated.

We have deleted duplicate references 12 and 21.

The citations 15 and 17 on line 26 do not align with the context of the article.

We have removed references 15 and 17 that are inconsistent with the context.

The citation 31 in line 84 does not correspond to the content of the article.

We have replaced citation 31 with citation 40 in line 84.

*[40] Bisht, R., Chattopadhyay, B. Formal Ir-catalyzed ligand-enabled ortho and meta borylation of aromatic aldehydes via in situ-generated imines. J. Am. Chem. Soc. **138**, 84-87 (2016).*

The reference 31 in line 144 should be updated to reference 28.

We have updated reference 31 in line 144 to reference 28.

*Hoque, M. E., Bisht, R., Haldar, C. & Chattopadhyay, B. Noncovalent interactions in Ir-catalyzed C-H activation: L-shaped ligand for para-selective borylation of aromatic esters. J. Am. Chem. Soc. **139**, 7745-7748 (2017).*

The author of line 149 should be revised to Itami and Chattopadhyay.

In 2019, the Phipps and Smith, Maleczka group reported the para-borylation of phenol and benzyl alcohol derivatives using steric and ion-pair directed strategies^{29,30}.

Quote 27 on line 171 also does not align with the context here.

We have revised reference 27 on line 171 to reference 32.

*[32] Saito, Y., Segawa Y. & Itami K. para-C-H borylation of benzene derivatives by a bulky iridium catalyst. J. Am. Chem. Soc. **137**, 5193-5198 (2015).*

(6) The substrate number in line 221 of the main text should be changed from 102 to 101 to match the number in Table 5.

Response: *Thanks for the reviewers for taking precious time to check the manuscript. We have corrected this mistake.*

*"the 2-(2-methyl-1,3-dioxolan-2-yl)phenol (**101**) and 1-(2-methoxyphenyl)ethan-1-one oxime (**103**) substrates delivered the desired para-borylated products (**102** and **104**)."*

(7) In the spectrum annotation on page 5 of SI, "meat" should be corrected to "meta". The NMR data for substrates 1a, 1b, 30a, 45f, and products 3, 4, etc. contain errors. Please thoroughly review all data and descriptions in the SI to ensure the accuracy of this research.

Response: *Thanks for the reviewers for taking precious time to check the SI. We carefully checked the SI and corrected the corresponding mistakes. At the same*

time, other NMR data errors and numbering errors were also corrected.

In the spectrum annotation on page 5 of SI, "meat" should be corrected to "meta".

We have corrected "meat" to "meta".

The NMR data for substrates **1a**, **1b**, **30a**, **45f**, and products **3**, **4**, etc. contain errors.

We have made corresponding revisions to the NMR data errors for substrates **1a**, **1b**, **30a**, **45f**, and products **3**, **4**, etc.

2-chlorobenzaldehyde O-(tert-butyl dimethylsilyl) oxime (**1a**)

$^1\text{H NMR}$ (400 MHz, Chloroform- d) δ 8.63 (s, 1H), 7.91 (dd, $J = 7.7, 1.8$ Hz, 1H), 7.37 (dd, $J = 7.9, 1.5$ Hz, 1H), 7.32 – 7.23 (m, 2H), 1.00 (s, 9H), 0.26 (s, 6H). $^{13}\text{C NMR}$ (101 MHz, CDCl_3) δ 150.4, 133.9, 130.8, 130.6, 130.0, 127.3, 127.0, 26.3, 18.4, -5.1. **HRMS (ESI)** m/z Calcd for $\text{C}_{13}\text{H}_{20}\text{ClNNaOSi}$ [$\text{M}+\text{Na}$] $^+$ 276.0946, Found: 276.0952.

2-chlorobenzaldehyde O-triethylsilyl oxime (**1b**)

The product was obtained as a colorless oil. $^1\text{H NMR}$ (400 MHz, CDCl_3) δ 8.61 (s, 1H), 7.90 (dd, $J = 7.6, 2.0$ Hz, 1H), 7.43 – 7.35 (m, 1H), 7.34 – 7.22 (m, 2H), 1.54 – 1.40 (m, 6H), 1.00 (t, $J = 7.3$ Hz, 9H), 0.83 – 0.73 (m, 6H). $^{13}\text{C NMR}$ (101 MHz, CDCl_3) δ 150.3, 133.9, 130.8, 130.7, 123.0, 127.3, 127.0, 18.4, 16.8, 16.3. **HRMS**

(ESI) m/z Calcd for $C_{13}H_{20}ClNNaOSi$ $[M+Na]^+$ 276.0946, Found: 276.0955.

3,4-dihydronaphthalen-1(2H)-one O-(tris((2,6-dimethylheptan-4-yl)oxy)silyl) oxime (30a)

1H NMR (400 MHz, $CDCl_3$) δ 8.05 (d, $J = 7.9$ Hz, 1H), 7.26 (td, $J = 7.3, 1.4$ Hz, 1H), 7.21 – 7.11 (m, 2H), 4.22 – 4.12 (m, 3H), 2.83 (t, $J = 6.7$ Hz, 2H), 2.76 (t, $J = 6.0$ Hz, 2H), 1.91 – 1.79 (m, 8H), 1.57 – 1.47 (m, 6H), 1.26 (ddd, $J = 13.4, 8.1, 5.2$ Hz, 8H), 0.88 (dd, $J = 7.9, 6.6$ Hz, 36H). ^{13}C NMR (101 MHz, $CDCl_3$) δ 158.9, 139.8, 131.0, 129.2, 128.5, 126.1, 125.0, 70.6, 46.9, 30.1, 24.7, 24.4, 23.3, 22.8, 21.7. HRMS (ESI) m/z Calcd for $C_{37}H_{67}NNaO_4Si$ $[M+Na]^+$ 640.4732, Found: 640.4750.

2,3-difluorobenzyl tris(2,6-dimethylheptan-4-yl) silicate (45f)

1H NMR (400 MHz, $CDCl_3$) δ 7.32 – 7.24 (m, 1H), 7.11 – 6.98 (m, 2H), 4.93 (s, 2H), 4.13 – 4.02 (m, 3H), 1.82 – 1.70 (m, 6H), 1.50 – 1.39 (m, 6H), 1.26 (ddd, $J = 13.5, 7.9, 5.4$ Hz, 6H), 0.90 – 0.83 (m, 1H). ^{13}C NMR (101 MHz, $CDCl_3$) δ 150.4 (dd, $J = 247.0, 12.4$ Hz), 146.7 (d, $J = 12.9$ Hz), 130.6 (d, $J = 11.1$ Hz), 123.8 (dd, $J = 6.8, 4.6$ Hz), 123.6 (t, $J = 3.4$ Hz), 115.7 (d, $J = 17.1$ Hz), 70.4, 58.8 (dd, $J = 5.1, 3.1$ Hz), 46.9, 24.6, 23.2, 22.7. ^{19}F NMR (377 MHz, $CDCl_3$) δ -140.0 (d, $J = 20.9$ Hz), -145.1 (d, $J = 21.1$ Hz). HRMS (ESI) m/z Calcd for $C_{34}H_{62}F_2NaO_4Si$ $[M+Na]^+$ 623.4278, Found: 623.4283.

2-methoxy-4-(4,4,6,6-tetramethyl-1,3,2-dioxaborinan-2-yl)benzaldehyde O-(tris((2,6-dimethylheptan-4-yl)oxy)silyl) oxime (3)

1H NMR (400 MHz, $CDCl_3$) δ 8.61 (s, 1H), 7.82 (d, $J = 7.7$ Hz, 1H), 7.52 – 7.34 (m, 2H), 4.26 – 4.10 (m, 3H), 3.90 (s, 3H), 1.93 (s, 2H), 1.91 – 1.74 (m, 6H), 1.51 (ddd, $J = 13.6, 7.6, 6.0$ Hz, 6H), 1.44 (s, 12H), 1.26 (ddd, $J = 13.5, 8.0, 5.2$ Hz, 6H), 0.89 (dd, $J = 8.6, 6.7$ Hz, 36H). ^{13}C NMR (101 MHz, $CDCl_3$) δ 157.1, 150.4, 126.2, 125.6, 122.8, 116.1, 71.2, 70.6, 55.8, 49.2, 46.9, 32.0, 24.4, 23.3, 22.8. ^{11}B NMR (128 MHz, $CDCl_3$) δ 26.2. HRMS (ESI) m/z Calcd for $C_{42}H_{78}BNNaO_7Si$ $[M+Na]^+$ 770.5533, Found: 770.5541.

2-ethoxy-4-(4,4,6,6-tetramethyl-1,3,2-dioxaborinan-2-yl)benzaldehyde O-(tris((2,6-dimethylheptan-4-yl)oxy)silyl) oxime (4)

1H NMR (400 MHz, $CDCl_3$) δ 8.65 (s, 1H), 7.82 (d, $J = 7.7$ Hz, 1H), 7.39 (d, $J = 7.7$ Hz, 1H), 7.34 (s, 1H), 4.28 – 4.07 (m, 5H), 1.92 (s, 2H), 1.91 – 1.77 (m, 6H), 1.57 – 1.46 (m, 6H), 1.45 – 1.41 (m, 15H), 1.31 – 1.20 (m, 6H), 0.89 (dd, $J = 8.8, 6.5$ Hz, 36H). ^{13}C NMR (101 MHz, $CDCl_3$) δ 156.6, 150.7, 126.1, 125.5, 122.9, 117.3, 71.2, 70.6, 64.2, 49.2, 46.9, 32.0, 24.4, 23.3, 22.8, 15.0. ^{11}B NMR (128 MHz, $CDCl_3$) δ 26.3. HRMS (ESI) m/z Calcd for $C_{43}H_{81}BNO_7Si$ $[M+H]^+$ 762.5807, Found: 762.5815.

Reviewer #2 (Remarks to the Author):

The manuscript describes para-selective Ir-catalyzed C-H borylation of arenes bearing bulky silyl protecting groups. The work allows the control of para-selectivity across a wide range of substrates including silyl oximes from aldehydes and ketones, silyl esters of carboxylic acids, silyl ethers of benzyl alcohols and phenols, and aryl silanes, taking advantage of steric hindrance. While the work is highly valuable in the context of organic synthesis, it lacks conceptual novelty as it relies on the strategy of steric hindrance between the Ir catalyst and substrate substituents pioneered by Itami et al. As such, the paper represents a logical extension and improvement of the known methods and may not be considered highly significant in the broader scientific context. Therefore, the reviewer recommends its publication in a specialized journal

Response: We thank the reviewers for their comments, but we believe that our work has sufficient conceptual novelty. We therefore list here sufficient reasons to demonstrate the conceptual novelty of our work.

*Steric hindrance effects (J. Am. Chem. Soc. **2015**, 137, 5193–5198; Angew. Chem. Int. Ed. **2022**, **61**, e202203539; Science **2022**, 375, 658-663;) and weak interactions (Sci. Adv. **2023**, 9, eadg3311; Chem. Soc. Rev. **2022**, 51, 5042-5100) are the main approaches in the field of iridium-catalyzed remote C-H borylation of aromatics to achieve the site-selectivity. Controlling site selectivity through modification of ligands (catalyst engineering) and substrates (catalyst engineering) is currently the mainstream strategy in this field. In the Ir-catalyzed remote C-H borylation of arenes to date, strategies to achieve meta-C-H borylation have emerged in an endless stream (Nat. Commun. **2023**, **14**, 6906), but there are relatively few reports on para-C-H borylation, especially in general and universal strategies.*

In this manuscript, we first disclosed a new silicon-protecting agent to perform para-selective C-H borylation reaction of various arenes. This protocol can directly use the classic commercial phenanthroline ligands in borylation without the need for the complex design of the ligands and has unprecedented substrate adaptability. Importantly the reaction all performed well under air in rather a short reaction time. This manuscript includes many new developments in C-H borylation reactions as listed below:

(1) The para-selective C-H borylation of aromatic ketones derivatives were first reported.

(2) Para-selective C-H borylation of aryl and benzyl silanes were first disclosed, leading to various novel B,Si bimetallic reagents.

(3) The site-selective C-H borylation can be adjustable by installing the newly developed trialkoxysilane protecting group on different functional groups of one aromatic ring, which were firstly reported.

(4) Up to 131 substrates are reported, highlighting the synthetic utility of this new method.

(5) The strategy we developed cleverly utilizes synthetic novel silicon-protecting agents or phenylsilane, and the origin of para-selectivity is fully governed by the steric crowding between the silicon protecting-group of the substrate and the dual steric hindrance effect of the borylation reagent.

In summary, we believe that our strategy is conceptually novel and effectively promotes progress in the field of C-H borylation.

Reviewer #3 (Remarks to the Author):

This manuscript reports on the development of para-Selective C–H Bond Borylation of Arenes by Trialkoxysilane-Induced Iridium-Catalyzed. An ideal approach for the construction of aryl boron compounds is to selectively replace a C–H bond in arenes with a C–B bond. However, controlling regioselectivity is one of the most challenging aspects of these transformations due to subtle differences in the reactivity of diverse C–H bonds. This novel trialkoxysilane reagent enables the iridium-catalyzed selective para-C-H boronization of various functionalized aromatic rings without inert gas protection. Some basic aromatic ketone and aromatic aldehyde derivatives can also be successfully applied to the reaction. Para-selective C–H borylation of aryl and benzyl silanes was also achieved, leading to various B,Si bimetallic reagents. Through their strategy, the authors achieved a series of late boronic modifications of drugs, demonstrating the application of their reactions. Mechanistic and computational studies revealed that the steric hindrance of the trialkoxysilane protecting group plays a key role in dictating the para-selectivity. Accordingly, I am in favor of publication in Nature Communications upon a couple of issues that need to be addressed.

Response: *We thank the reviewer for the positive comments.*

(1) In Figure 1b, the challenge described by the authors is that of para-C-H borylation of aromatic ketones, aromatic aldehydes, however, there is an ambiguity in expression in terms of the authors' substrate studies using derivatives of aromatic ketones and directional aldehydes.

Response: *We thank the reviewer's valuable advice. We have revised Figure 1B to highlight the relationship between aldehyde ketones and aldehyde ketone oxime derivatives. Similarly, we have made corresponding modifications to the manuscript.*

“However, ketone, aldehyde oxime derivatives, benzylsilanes, and multi-functionalized substrates have still not been reported for C–H borylation,”

(2) The condition screening is suggested to be added in supplementary materials.

Response: We thank the reviewer's valuable advice. We have added conditional screening and corresponding ^1H NMR in the supplementary materials.

2.1 Effects of various silicon functional groups, ligand and temperature for *para* C-H borylation^a

^aReaction conditions: substrate **1** (0.2 mmol) B_2dmg_2 (1.5 equiv), $[\text{Ir}(\text{OMe})\text{cod}]_2$ (1.5 mol%), Me_4phen (3.0 mol%), cyclohexane (1 mL), 100 °C, 1 h, isolated yield. Ratios of meta to para were determined from the crude ^1H NMR spectra after borylation. ^b B_2pin_2 used instead of B_2dmg_2 .

2.2 Effects of various silicon functional groups for *para*-selective C-H borylation^a

^aSubstrate **1**, 0.2 mmol, B₂dmg₂ (1.5 equiv), [Ir(OMe)cod]₂ (1.5 mol%), and Me₄phen (3.0 mol%), cyclohexane (0.2 M), 100 °C, 30 min, isolated yield. ^bReaction scale 3.0 mmol. Ratios of *meta* to *para* were determined from the crude ¹H-NMR spectra after borylation.

Typical Procedure for the Silylation of Arenes:

Arylsilane **4-1** (0.2 mmol, 1.0 equiv), B₂dmg₂ (84.6 mg, 0.3 mmol, 1.5 equiv), [Ir(cod)OMe]₂ (2.0 mg, 0.003 mmol), Me₄phen (1.5 mg, 0.006 mmol), and Cyclohexane (1.0 mL) were added to a 15 mL glass vial under air atmosphere. The glass vial was capped with a teflon pressure cap and placed into an aluminum block pre-heated to 100 °C for 30 minutes. After completion, cyclohexane was removed under reduced pressure and chromatographic separation with silica gel (20% ethylacetate in hexane as eluent) gave the borylated product **4-2** as colorless liquid. The *para:meta* ratio of products was reported from analysis of ¹H NMR.

2.3 Effects of various boron reagents for *para*-selective C-H borylation^a

^aSubstrate **4-1**, 0.2 mmol, boron reagent (1.5 equiv), $[\text{Ir}(\text{OMe})\text{cod}]_2$ (1.5 mol%), and Me_4phen (3.0 mol%), cyclohexane (0.2 M), 100 °C, 30 min, isolated yield. ^bReaction scale 3.0 mmol. Ratios of *meta* to *para* were determined from the crude ¹H-NMR spectra after borylation.

(3) Boc protecting group should be deprotected to form free amine on Table 3.

Response: We thank the reviewer's valuable advice. We conducted a gram scale reaction and also deprotected the Boc and silyloxy group of product **58**.

“Similarly, the substrate **58a** derived from the conversion of 2-hydroxybenzylamine underwent a gram scale reaction (**58**) and subsequent deprotection (**156**), resulting in good yield and selectivity. To our delight, Boc and silyloxy group can be easily removed simultaneously in the presence of trifluoroacetic acid.”

(4) The authors should have tried the substrate part with a heterocyclic compound.

Response: We thank the reviewer's valuable advice. We attempted to use thiophene, furan, and pyridine heterocyclic compounds as matrix components. Although the yield of C-H borylation is good, unfortunately, the site selectivity is poor, and only mixed products can be obtained. We speculate that this is due to the spatial hindrance of the methylsioxane protecting group not effectively covering the C-H bonds on both sides of the heterocycle, resulting in a decrease in site selectivity.

Reviewers' Comments:

Reviewer #1:

Remarks to the Author:

The authors appropriately addressed the reviewers' comments. The manuscript is suitable for publication in its current form.